

# Vertical and lateral manipulation of single Cs atoms on the semiconductor InAs(111)A

Rian A. M. Ligthart[1], Cristophe Coinon[2], Ludovic Desplanque[2],
Xavier Wallart[2] and Ingmar Swart[1*]

**1** Debye Institute for Nanomaterials Science, Utrecht University, the Netherlands
**2** Univ. Lille, CNRS, Centrale Lille, Univ. Polytechnique Hauts-de-France, UMR 8520 - IEMN - Institut d'Electronique de Microélectronique de Nanotechnologie, F-59000 Lille, France

⋆ I.Swart@uu.nl

## Abstract

The tip of the scanning tunneling microscope can be used to position atoms and molecules on surfaces with atomic scale precision. Here, we report the controlled vertical and lateral manipulation of single Cs atoms on the InAs(111)A surface. The Cs adatom adsorbs on the In-vacancy site of the InAs(111)A—(2x2) surface reconstruction. Lateral manipulation is possible in all directions over the surface, not just along high-symmetry directions. Both pushing and pulling modes were observed in the height profile of the tip. We assembled two artificial structures, demonstrating the reliability of the manipulation procedures. Structures remained intact to a temperature of *at least* 44 K.

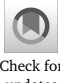

# 1 Introduction

Structures created atom-by-atom have been used to push the limits of information processing, information storage, and further our understanding of materials. [1–7] These atomically well-defined assemblies are typically made using the tip of the scanning tunneling microscope (STM). Single adatom manipulations are useful to study adsorption geometries, single molecule chemistry [8,9], switching events [10,11], spin lattices [12,13] and the confinement of surface state electrons which can create new band structures. [14–16]

Manipulation in an STM can be divided into two categories, lateral and vertical manipulation. When manipulating laterally, the adatom is either pulled (pushed) across the surface due to an attractive (repulsive) tip-adsorbate interaction. [17,18] The adatom remains on the surface; the diffusion barrier to lateral motion is overcome by the tip-adsorbate interaction. This mode is typically used on metallic substrates, where the potential landscape is rather flat. Secondly, vertical manipulation is the reversible transfer of the adsorbate between tip and surface. For In atoms on In/As(111), this transfer of the adatom *from* the surface to the tip is mediated by a tip-induced electric field and vibrations that arise due to inelastically tunelling electrons. The transfer from the tip *towards* the surface is in this case dependent on short-range forces such as Van der Waals forces and is achieved by an almost point contact between tip and sample. [19] Vertical manipulation facilitates faster adatom manipulation over longer distances (>20 nm) since the adsorbate is carried on the tip. Lateral manipulation, on the other hand, is considered more stable since the tip apex does not change with each manipulation (which hampers imaging and consequently the accuracy with which atoms/molecules are deposited). In most cases, manipulation experiments were performed on metallic substrates, where the potential energy landscape is relatively flat. However, it is possible to perform manipulations on insulating films and semiconductor surfaces. [19,20] For the latter, atoms could be pulled out of the surface or vacancy sites could be moved over the surface. [21–26] Furthermore, substitutions of single impurities into several III-V semiconductors have been made. [27–29] In all these cases, vertical manipulation or inelastic excitation was used.

Recently, manipulations of adatoms on top of semiconductors were used to construct artificial electronic lattices. [7,30] An advantage of using semiconductors over metallic substrates is that the former allow for higher energy resolution of the electronic states of the artificial lattices. Lattices constructed using CO/Cu(111) make use of the 2D electron gas (surface state) of the Cu(111) substrate. However, CO molecules on the surface can also scatter the surface state electrons to bulk states of the Cu(111), thereby reducing the lifetime and causing significant broadening of the confined states . [6] In the case of semiconductors, the charged adatoms induce states in the band gap of the semiconductor due to local band bending. Consequently, there are no bulk states available to scatter to which leads to much narrower linewidths for confined states on semiconductors compared to metals. However, because of the more corrugated potential energy landscape, atomically precise manipulations on semiconductor surfaces are more demanding than on metallic substrates. For successful lateral manipulation the tip-adsorbate interaction needs to be stronger than the adsorbate-substrate interaction and the barrier for lateral motion should be smaller than for desorption.

The In adatoms on InAs(111)A could be vertically manipulated to create quantum dot like structures, as well as an SSH chain [19,31] whereas on InSb(110), Cs was manipulated laterally to simulate molecular orbitals. [7] The more polar and covalent character of the III-V semiconductor make manipulations on top of a surface more demanding in comparison with a metallic substrate. However, the larger radius of the Cs compared to the substrate atoms is expected to facilitate lateral manipulation. Here, we report the controlled lateral and vertical manipulation of Cs on InAs(111)A. The adsorption geometry of Cs on InAs(111)A is determined and the mechanism responsible for the lateral manipulation is established. Finally,

the reliability of the manipulation methods is demonstrated by the construction of artificial nanostructures built with atomic precision. The structures were observed to be stable until a temperature of *at least* 44 K. Due to an unintentional tip-crash, we do not have data for higher temperatures. However, individual Cs atoms remained stable at 77 K, suggesting a higher stability for artificial structures.

## 2 Methods

The STM experiments were performed using a Scienta Omicron LT-STM set-up with a base temperature of 4.3 K and a pressure in the $10^{-10}$ mbar range. An In-coated PtIr tip, possibly coated by some Cs atoms, was used for the measurements. The tip was coated with In using the following procedure. First, the tip was brought close to the surface (typical parameters: 500 mV, 300 pA). Then, we turned off the feedback loop and applied a bias of 10 V for a few seconds. This locally melted the surface, causing the formation of an In cluster. Small tip indentations in these clusters result in an In coated PtIr tip.

A 100 nm thick layer of InAs was grown on top of an InAs(111)A crystal (acquired from SWI, n-doped with a concentration between $1x10^{17}$ - $3x10^{18}$ cm$^{-3}$) by MBE in a RIBER Compact21 system. The surface is deoxidized by a first annealing under a combination of a hydrogen atomic flux and an As ($As_2$ or $As_4$) one up to 440 °C and then by a final annealing to 480 - 490 °C under an As flux only. It results in a sharp 2x2 Reflection High Energy Electron Diffraction (RHEED) pattern. 100 nm InAs is then grown at 470-480 °C at a growth rate of 0.15 - 0.2 monolayer/s and an In/As flux ratio between 30 and 40 where the In and As fluxes were determined by RHEED intensity oscillations. The substrate temperature was measured by an infrared optical pyrometer calibrated against the InSb melting point at 525 °C. During growth, the RHEED pattern remains 2x2 whereas the resulting surface RMS roughness determined by atomic force microscopy (AFM) on 1x1 $\mu m^2$ images after growth lies in the 0.2-0.3 nm range. After MBE growth, the sample was cooled down to a temperature below 20 °C for the deposition of an amorphous As capping layer, a few tenths of nm thick. This latter was removed inside the preparation vacuum chamber of the LT-STM by annealing the crystal at 552 K for 1 hour. Caesium was evaporated from a $Cs_2CrO_4$ filament (SAES Getters) on to the InAs(111)A surface at 4 K. Scanning tunelling spectroscopy was performed by applying an ac-modulation (at 769 Hz) of 1 mV (rms amplitude) to the bias signal and using a lock-in amplifier for detection. STM images were processed with Gwyddion 2.56, a plane subtraction and 2D-FFT filtering were performed for clarity. The schematics were prepared with Blender software Version 2.93.5.

## 3 Results and discussion

### 3.1 Cs/InAs(111)A

Figure 1a shows an STM topograph of the InAs(111)A surface with single Cs and In atoms (orange and green circles, respectively). The In atoms are naturally present on the InAs(111)A surface and adsorb on the In vacancy sites. [19, 32] The Cs atoms are depicted as bright protrusions and do not have the characteristic depletion ring as observed for the adsorbed In. In addition, also a domain wall (orange arrow) and a hole are seen, plus some unknown aggregates. The InAs(111)A is In terminated and has a 2x2 surface reconstruction. [32] The surface reconstruction creates a hexagonal lattice of In vacancy sites. Figure 1b shows the surface reconstruction and demonstrates that the reconstruction is unaffected by the deposition of the

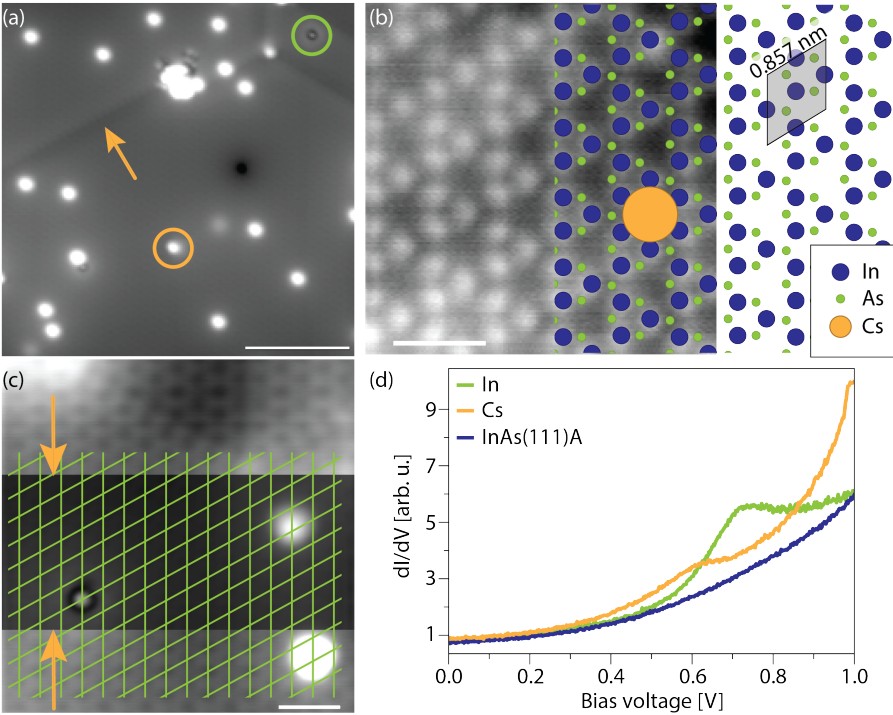

Figure 1: (a) STM constant current topography showing Cs atoms (orange circle), In atom (green circle) and a domain wall (orange arrow) on the InAs(111)A surface. Image is 35x35 nm - scale bar is 10 nm - and taken at 0.8 V and 20 pA. (b) STM constant current image showing the single In atoms with overlay of the surface reconstruction of InAs(111) with unit cell of 0.857x0.857 nm and Cs adatom (orange). Image is 4x4 nm - scale bar is 1 nm - and taken at 0.5 V and 50 pA. (c) Constant current image of two single Cs atoms and one In atom with a grid overlay to overlap with the In vacancy sites to show the adsorption geometry of the Cs and In adatoms. At the top of the image a domain wall is present. Scale bar is 2 nm, image taken at 100 mV and 30 pA. (d) Averaged dI/dV spectra taken on top of a single Cs atom, single In atom and on InAs(111)A background.

**Cs atoms.** We determine the adsorption site of the Cs atoms by using the lattice of vacancy sites as a reference. Figure 1c shows an STM topograph of an area with one In and two Cs adatoms. At the lines indicated by the orange arrows, the contrast was adjusted so that the vacancy lattice (bottom and top part of the image) and In and Cs adatoms are optimally visible. The intersections of the green lines correspond to vacancy sites. We find that the centres of the In atom, as well as that of the two Cs atoms are located directly above a vacancy site. Hence, Cs adsorbs on the In vacancy site position. To confirm that the adsorbate is electronically different from the In adatoms, dI/dV spectroscopy was performed on top of the Cs and In atoms, as well as on the bare InAs(111)A surface. The spectrum on the In adatom, green curve in Figure 1d, shows the known electronic state at 680 mV which is attributed to an unoccupied state from the atomic orbitals of In. [33] For the Cs atom (orange curve), there is a shoulder around 600 mV. Given the low ionization energy of Cs, and its behaviour on InSb(110), the Cs atoms on InAs(111)A are expected to be charged. This is confirmed by the observation of a localized electronic state for a cluster of Cs atoms (see below).

## 3.2 Vertical manipulation

Since vertical manipulation is the dominant method used on semiconductors, we first focus on this technique. The procedure is illustrated in Figure 2a: (i) the feedback loop is interrupted and the bias voltage is increased (ii) the tip is brought closer to the adatom until the adatom jumps to the tip, (iii) the tip, with the atom attached to the apex, is moved to the desired location, (iv) the tip is brought closer to the surface so a new bond between the adatom and the surface can be formed. For Cs on InAs(111)A, we find that the following procedure works well: to pick up the atom, the tip was taken out of feedback (0.5 V, 30 pA) and the bias voltage was increased to 1.5 V. Bringing the tip closer to the surface by approximately -0.46 to -0.56 nm, results in a sudden decrease in the current, signalling the transfer of the atom from the substrate to the tip. Typically, the current exceeds 300 nA. The vertical manipulation procedure for Cs atoms on InAs(111)A, as well as the experimental signatures, are similar to the case of In adatoms. This suggests that the manipulation mechanisms are similar for the two different atoms, i.e. that jump of the Cs adatom is caused by inelastically tunelling electrons which excite vibrations. [19]. This is consistent with the observation that the likelihood of picking up an adatom is larger at higher tunelling currents. The picking up of a single Cs atom is presented in Figure 2b and 2c. After the transfer of Cs, the tip can continue to scan the surface. Upon picking up a Cs atom, there can be minimal changes in how the tip images the surface. For example, adsorbates can look more or less circular or there is a slight offset in position. With a stable tip, the tip apex returns to its previous shape again after the Cs is put down. This behaviour is different from the vertical manipulation of In atoms on InAs(111)A, which requires an In terminated tip with a specific apex amenable to pick up In atoms (pocket of two or three atoms). The Cs is transferred back to the substrate by bringing the tip closer to the surface, by changing the feedback settings to 300 pA and 10 mV. The Cs – tip bond is broken and a new one between the tip and the surface is formed. Putting the adatom down is not induced by electric field but mediated by a near point-contact between the tip and the adatom.

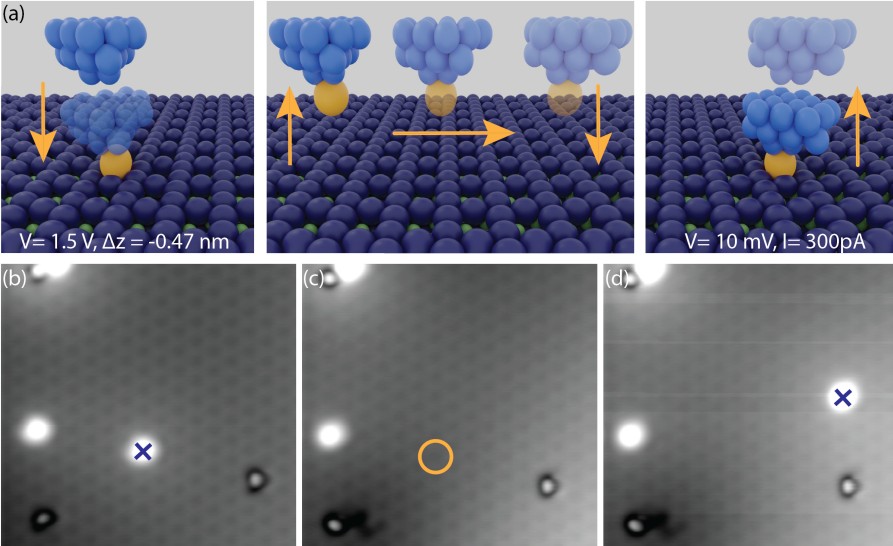

Figure 2: (a) Schematic of vertical manipulation, showing the picking up, the moving of the adatom on the tip and the putting down of the atom. (b) - (d) Constant current image showing the picking up of the Cs adatom (marked by the blue cross), the missing of the picked up Cs atom (marked by the orange circle) and the putting down of the Cs atom (marked by the blue cross). STM image 12x12 nm - scale bar is 2 nm - taken at 100 mV and 30 pA.

As Cs atoms could be transferred back to the surface at various positive and negative voltages, the deposition process is not induced by the electric field, but is mediated by the near point contact between tip and adatom. We found that when the tip would have a perfectly centred In atom on its apex, Cs atoms could not be picked up. After depositing the In on the substrate using a similar procedure as just described for the Cs atom, vertical manipulation of Cs was possible again. Furthermore, it is possible to controllably pick up multiple Cs atoms. However, transferring a second atom to the tip required the tip to be brought closer to the surface than for the first pickup. This implies a repulsion between the Cs atoms, but also between the Cs and the In atoms. Since the vertical manipulation procedures for Cs and In on InAs(111)A are similar, and the latter has been investigated in detail [19], we did not investigate this method further.

## 3.3 Lateral manipulation

As we will now show, Cs atoms can also be manipulated laterally. Lateral manipulation has the advantage that in principle the tip does not change during the process, making the assembly of nanostructures easier to track. The general procedure follows the same steps as for metallic substrates, see Figure 3a: (i) lowering the tip above the adatom, (ii) moving the tip at the same height along the surface capturing the adatom underneath and (iii) retracting the tip to the original height releasing the adatom, see the schematic in Figure 3a. During this process, the bond between the Cs and the InAs(111)A surface is largely preserved. The Cs atom on InAs(111)A is manipulated by lowering the tip above the Cs atom to a feedback value of 10 mV and 300 pA and subsequently moving the tip along the surface. A controlled manipulation of a single Cs atom along the closed packed direction of the In vacancies is shown in Figure 3b and 3c. The Cs could also be manipulated in non-closed packed directions and over distances exceeding 6 nm.

The Cs is moved over the surface by a potential well created by the tip which captures the adatom. [34] If the tip-adsorbate interaction is strong enough to overcome the diffusion barrier, the adatom is either pulled/ slid/ pushed along the surface as the tip moves. To establish which

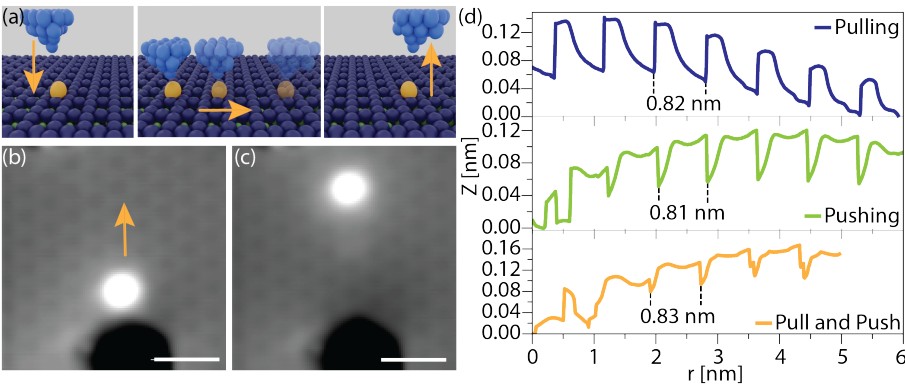

Figure 3: (a) Schematic of lateral manipulation where the tip is moved closer to the adatom after which it moves in a lateral direction and pulls/ slides/ pushes the adatom with it, finally, the tip is retracted again to scan height. (b) - (c) Constant current image showing the lateral movement of a Cs atom along a closed packed row of the InAs(111)A surface in the direction of the orange arrow. STM image taken at 500 mV and 50 pA with a scale bar of 2 nm. (d) Profile of relative tip height (Z) versus length of manipulation (r) showing the lateral manipulation over a closed packed row of pulling (blue line), pushing (green line) and a pull which continues in pushing (orange line). The averaged distance between the hops are indicated.

process occurs for Cs atoms on InAs(111)A, traces of tip height versus manipulation length along a high-symmetry direction of the surface were recorded, see Figure 3c. We observe two types of curves, shown in blue and green. For the blue curve, the tip gradually moves towards the surface, followed by a sudden retraction. The green curve shows the opposite behaviour: a gradual increase, a plateau, and sudden decrease in tip-sample distance. In both cases, the lateral distance between the sudden jumps is equal to the measured distance between vacancy sites. These two curves are indicative of pulling and pushing, respectively. [17] First focus on the blue curve. The tip scans over the atom and as it moves laterally to the next adsorption site, it approaches the surface (tip is in feedback), explaining the gradual decrease in tip height. Just before it reaches the next stable adsorption site of the Cs atom, the tip suddenly retracts, signalling that the Cs atom has jumped to the vacancy site underneath the tip. For the green curve, the situation is opposite: as the tip laterally approaches the Cs atom, the tip-height gradually increases. Before reaching the maximum, the atom is pushed away to a nearby adsorption site, causing a sudden decrease in tip-sample distance. Interestingly, the same tip can exhibit pushing and pulling behaviour when moving in opposite directions. We observed pushing characteristics along a crystallographic direction. Upon moving the same atom back to its original position along the same trajectory using the same tip, pulling motion was observed. This suggests that the mesoscopic tip shape influences the pulling or pushing behaviour of the tip. Interestingly, we sometimes observed pushing and pulling in one trace, see the bottom panel of Figure 3d, where the adatom is pulled for one hop and then pushed further. Depending on the tip apex, the Z(r) profile can look more or less pronounced.

We have investigated the reliability of the lateral manipulation by recording the success rate as a function of setpoint current. Using one tip apex, 120 manipulations were performed. Figure 4a shows a histogram of the results. The success rate increased from 0 % for I = 100 pA to 86 % for I = 500 pA. None of the manipulations at I = 100 pA were successful, while subsequent manipulations at I= 300 pA did still work, suggesting that a change in tip apex is not responsible. Increasing the setpoint current further, resulted in a transfer of the Cs atom to the tip instead of a lateral manipulation. We note that the success rate will vary somewhat from tip apex to tip apex.

The results described above suggest that it is possible to build nanostructures of Cs atoms on InAs(111)A with atomic scale precision. The clear visibility of the 2x2 surface reconstruction with its In-vacancy sites facilitates positioning atoms on desired locations. The spacing between the Cs adatoms is determined by the hexagonal symmetry of the vacancy reconstruc-

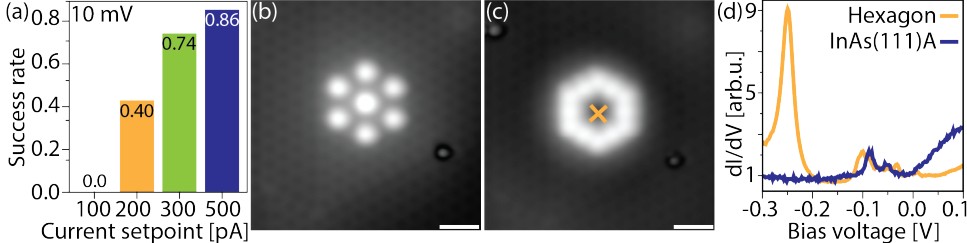

Figure 4: (a) The success rate of manipulations at 10 mV at different current setpoints: 100, 200, 300 and 500 pA. (b) Constant current image of 7 precisely located Cs atoms on the InAs(111)A surface. STM image of 12x12 nm - scale bar is 2 nm - taken at 0.1 V and 30 pA. (c) Constant current image of 12 precisely located Cs atoms in a closed packed hexagon configuration on the InAs(111)A surface. STM image of 12x12 nm - scale bar is 2 nm - taken at 0.1 V and 160 pA. The orange cross indicates the location of the dI/dV spectrum. (d) dI/dV spectra taken in the centre of the hexagon of (c) and on the clean InAs(111)A.

tion. We assembled a hexagon of Cs atoms with an additional atom in the centre, see Figure 4b. In this structure, the distance between two Cs atoms is two vacancy sites. It is also possible to position Cs atoms in neighbouring vacancy sites. Figure 4c shows a nanostructure where 12 Cs atoms were placed to form a hexagon with a width of 2.8 nm. The right and left wall of the hexagon are two non-closed packed In-vacancies apart. There was no sign of destabilization of the structure, despite of the Coulomb repulsion between the positively charged Cs atoms.

Bias spectroscopy was performed inside the hexagon shown in Figure 4c. Figure 4d shows the dI/dV curves acquired inside the hexagon and on the clean InAs(111)A. The spectrum of the hexagon exhibits a sharp peak at -0.25 V, demonstrating the formation of a localized electronic state inside the bandgap of the substrate. This verifies that the Cs atoms on InAs(111)A are charged and demonstrates that this platform can be used to construct artificial lattices. The full-width-at-half-maximum (FWHM) of the peak is 33 mV, much narrower than lineshapes observed for artificial lattices made using the CO/Cu(111) platform, but comparable to that of a 22-atom long In chain on InAs(111)A. [15,30] For In/InAs(111)A, the linewidth depends on the density of In atoms in the nanostructure (also expected for Cs/InAs(111)A) and can be as small as 10 mV for a line of six In atoms. [30] This is similar to the ≈7mV FWHM observed for the Cs octagon on InSb(110). [7]

## 3.4  Temperature

The stability of the nanostructures was tested by heating the sample. Figure 5a shows an STM topograph acquired at $T = 44$ K. The two nanostructures visible (two closed packed lines of 5 Cs atoms), built at $T = 4$ K, where stable to *at least* this temperature. This suggests that the diffusion barrier of the Cs atoms is not overcome at 44 K, since the Cs atoms did not start to hop. Upon further heating, the tip crashed and the stability of the structures above $T = 44$ K could not be studied further. Figure 5b and c show STM topographs of the same area, taken 33 hours apart. The sample temperature was $T = 77$ K. It is clear that at this temperature, there was no significant diffusion of Cs adatoms. This suggests (1) that artificial lattices could be assembled at this temperature and (2) that the structures could be stable at temperatures above $T = 44$ K, but further study is needed. The stability of the individual Cs atoms at $T = 77$ K could enable manipulations to be performed at higher temperatures.

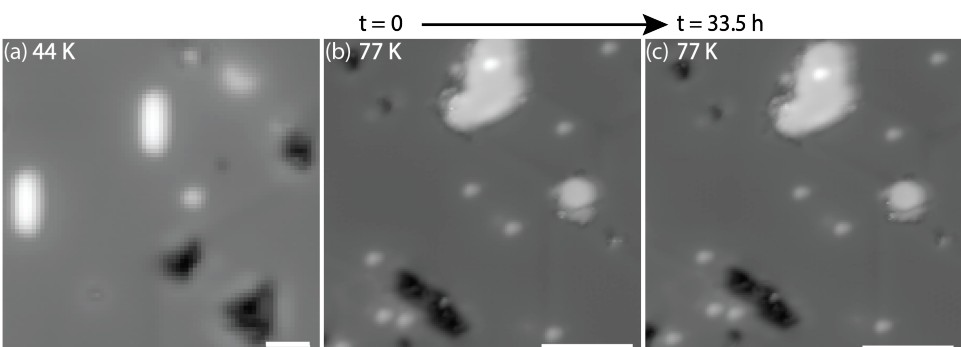

Figure 5: (a) Constant current image of two closed packed lines of 5 Cs atoms each. STM image taken at 44 K and at 900 mV and 30 pA, scale bar is 2 nm. (b) - (c) Constant current image of Cs on InAs(111)A at 77 K with a time difference of 33.5 h between (b) and (c). The Cs atoms remain on their adsorbed sites and no observations of hops were made. Images of 35x35 nm taken at 900 mV and 30 pA.

## 4 Conclusion

In summary, we demonstrated the vertical and lateral manipulation of Cs atom on the InAs(111)A surface. The Cs atom appear as a protrusion adsorbed on an In-vacancy site, just as the native concentration of In adatoms does. The vertical manipulation of Cs atoms worked by picking up the Cs with a tip-induced electric field whereas the putting down was governed by short-range forces. The lateral manipulation of Cs atoms could be achieved by bringing the tip closer to the surface by changing the feedback settings to 300 pA and 10 mV. During manipulation, both the pushing and pulling of Cs atoms were observed due to attractive/ negative tip – Cs interactions. The success rate of the manipulation depended on the current setpoint, and can be as high as 86 %. To demonstrate the atomic precision and the reliability of the Cs manipulation, a closed packed Cs hexagon was built. The artificial structures remained intact to at least $T = 44$ K and individual Cs atoms did not diffuse at $T = 77$ K. The controlled manipulation of Cs atoms on InAs(111)A broadens the opportunity to build artificial structures with atomic precision on a semiconductor.

## Acknowledgments

We thank D. Vanmaekelbergh for useful discussions.

**Author contributions** RL performed the experiments and data analysis. MBE-growth was done by CC, LD, XW. IS conceived and supervised the research. All authors contributed to writing the manuscript.

**Funding information** The research was made possible with financial support from the European Research Council (Horizon 2020 "FRACTAL", 865570) and with the support of the IEMN CMNF platform of the French Technological Network RENATECH.

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
