# Peer review of "Vertical and Lateral Manipulation of Single Cs atoms on the Semiconductor InAs(111)A"

_SciPost Physics, doi:SciPost Phys. 16, 096 (2024)_

## Round 1 · Referee Report · Anonymous (Referee 1) · 2023-10-27

Report

Ligthart et al. present a study of a novel platform for artificial lattices consisting of individual Cs atoms on InAs(111)A. They report on different successful manipulation modes with the STM tip and present examples of small artificially constructed objects, including densely packed structures of Cs atoms. The topic is timely, the paper is well written and this is indeed an interesting development, so I would clearly recommend publication of this work after the authors commented on my questions below. Most importantly, I would like to see a little more data backing the claim that this platform is interesting for studies of artificial lattices.

  • The authors present a nice statistical evaluation of the manipulation's success rate in Fig. 4a (please add statistical error bars to Fig. 4a though). It would be interesting if they could provide similar statistics on the vertical manipulation mode.
  • On page 4, they state that stable tips does not change significantly after vertical manipulation, however, the tip in Fig. 2b-d becomes unstable after dropping the Cs atom. Do they have other examples for vertical manipulation with no clear tip changes?
  • In general, it would be interesting if the authors could comment on how they prepare the tip apex. Can you fix an unstable tip directly on the InAs surface? Are there band bending effects for different tip apexes?
  • In Fig. 3, it is shown that both pulling and pushing mode can be achieved. I am puzzled though what the difference is: does it mainly depend on the precise tip apex? Or is there a dependence on (e.g.) directionality of the lateral manipulation? Please clarify.
  • The authors write that semiconductors "allow for higher energy resolution in scanning tunnelling spectroscopy". I know what they want to express, but please rephrase this because it sounds as if the actual energy resolution improved, which is not true. Please also add one or two sentences about the physical reason for this improved "effective" resolution and the advantage of their platform over other experimental platforms for artificial lattices. This would highlight the work's quality and relevance throughout the paper.
  • They write: "Given the low ionization energy of Cs, and its behaviour on InSb(110), the Cs atoms on InAs(111)A are charged". I don't quite understand if this is a speculation or a backed claim? Please clarify.
  • Now that the authors showed that Cs can be moved, it would be important to provide more data showing that this platform actually qualifies for artificial lattice studies. For instance, is there any (sharp?) resonance peak around the structures presented in Fig. 4 verifying the confinement of charges?

---

## Round 1 · Referee Report · Philip Moriarty (Referee 2) · 2023-11-16

Strengths

  1. Clearly written paper on a subject of particular topical interest.

  2. Meets SciPost acceptance criterion: "Open a new pathway in an existing or a new research direction, with clear potential for multipronged follow-up work."

  3. Provides new protocols for single atom manipulation on an important substrate/platform .

Weaknesses

  1. Some claims re. manipulation mechanisms -- and, as pointed out by Reviewer #1, re. the charge state of the Cs adatoms -- do not seem to be fully supported by experimental evidence. See comments in report below.

  2. Not clear how the claimed indium termination of the tip is maintained throughout the experiments.

  3. Although the experimental measurements are generally described in more than adequate detail, in some places greater detail on error bars/uncertainties could be provided.

Report

This is an important and well-written paper on a topic of considerable interest to the nanoscience, surface science, and condensed matter physics communities: single atom manipulation on semiconductor surfaces. The Cs/III-V system is likely to receive renewed attention following the elegant and pioneering "quantum simulator" paper from the Radboud group (Khajetoorians et al.) published earlier this year (Science 380 1048 (2023)). Previous similarly elegant work by Folsch et al. [e.g. PRL 115, 076803 (2015)] on the assembly of quantum dot molecules on InAs(111)A has demonstrated that this particular surface is especially amenable to controlled single atom positioning with an STM tip.

As such, this paper is particularly timely and establishes protocols for manipulation of Cs atoms on InAs(111)A. I agree with the points raised by the first reviewer (i.e. Anonymous Report 1 on 2023-10-27) and would also ask for the following points to be addressed in a revised version of the manuscript:

1(a). The Introduction draws a distinction between vertical manipulation methods that transfer an atom from the tip to the substrate and vice versa, generalising that these mechanisms involve the electric field and inelastic tunnelling for surface-to-tip transfer, and vdW forces/short-range forces for tip-to-surface transfer. I think this is a rather strong statement if it is meant in a general sense. Are the authors suggesting that these mechanisms hold for the InAs(111)A substrate or do they mean that all vertical manipulation events, i.e. for any tip-sample combination, can be explained in this manner?

1(b). In a similar vein to 1(a) above, on p. 4 the authors argue that the jump of the adatom is "caused by inelastically tunnelling electrons which excite vibrations [20]." The reference is to a detailed analysis by F\"olsch's team that demonstrates that this is the case for In adatoms on InAs(111)A. It is not immediately obvious that the inelastic tunnelling mechanism will also dominate for the transfer of Cs adatoms. Do the authors have direct evidence for this inelastic tunnelling mechanism in their experiments on Cs/InAs(111)A?

1(c). The Introduction is a single, long paragraph. I suggest breaking this up into at least two shorter paragraphs to aid readability and clarity.

  1. The final sentence of the Introduction section is intriguing: "The structures remained stable until a temperature of at least 44 K, and at 77 K no hopping of individual Cs atoms was observed." This suggests that the barrier for diffusion for Cs atoms that comprise an STM-assembled structure is lower than that for individual Cs atoms. Am I interpreting this correctly?

  2. This point relates to the first reviewer's questions re. tip structure. In the second sentence of the Methods section, the authors state that an "In-coated PtIr tip was used for all measurements." (Emphasis mine.) How do the authors know that the tip is In-terminated, and, equally importantly, that it remains In-terminated throughout the manipulation experiments and spectroscopic probing? Is there not a finite (and perhaps rather large) probability of Cs being picked up by the tip?

  3. p.2, third line from the bottom: "...reflection high energy electron diffraction (RHEED)..."

  4. In Section 3.1, I suggest moving the sentence that starts "The In atoms are naturally present on the InAs(111)A surface..." so that it's the second sentence of that paragraph.

  5. First sentence in main text on p.4: "...results in a sudden jump down in the current". This should be "sudden increase in the current".

  6. p.5, second sentence. "Putting the adatom down is not induced by the electric field..." How do the authors rule out the influence of the electric field?

  7. Fig.3 -- The mean hopping distances in Fig. 3(d) are quoted to three significant figures (i.e. down to the single pm level). What's the standard error on these mean values?

  8. For Figure 4 (and in line with Reviewer 1's comment re. error bars), what was the total number of manipulation events used to generate the histogram? (My apologies if this information is in the paper and I missed it.) Is there a bias polarity dependence?

  9. Caption to Fig. 5, final sentence. I assume this should be "900 mV and 30 pA".

Requested changes

See above for numbered list.

---

## Round 1 · Referee Report · Anonymous (Referee 3) · 2023-12-8

Report

Ligthart et al. present a study on single Cs atom manipulation on semiconducting InSb(111)A surface including different manipulation modes. The paper is well written, clear for the reader and its strength lays in detailed description of lateral atom manipulation of adatoms on a semiconductor. I can recommend publication of this work after addressing the points listed below.

1) In the introduction authors state that the use of semiconductors as a substrate for creating artificial electronic states allows for higher energy resolution in STS. Do authors mean to say that use of a substrate directly influences STS resolution? Addressing the physical reason for sharper energy states obtained for atom assembled quantum dots on semiconducting substrates would be beneficial for clarifying the above statement.
2) Further in the introduction authors states that atom manipulation is more demanding on III-V semiconductor due to its polar and covalent character and I am missing a link. Can the authors elaborate on how exactly this character influences atom manipulation?
3) As a part of a motivation a part addressing the use of Cs atoms is missing. InAs(111)A substrate has native In adatoms which were previously used by e.g. Fölsch et al. [PRL 103, 096104 (2009) or Nature Nanotech 9, 505–508 (2014)]. What is the benefit of using Cs?
4) The vertical manipulation part of the paper is similar to the approach already described previously for In adatoms on the same substrate by Fölsch et al. [PRL 103, 096104 (2009)] but for with much less details. It would be nice to see a similar evaluation for Cs atom manipulation presented here especially that the lateral manipulation description is more detailed.
5) The authors state that the success rate for vertical manipulation “will vary somewhat” for different tip apexes based on data obtained for only single tip apex. What is the tip apex termination in this particular case? How does manipulation compare between bare PtIr, In terminated and Cs terminated tips?
6) The difference for data obtained for push and pull traces is not clear (Fig.3d). Are they obtained in different crystallographic directions or with different tip terminations? Is there any influence for manipulation based on local chemical environment (vacancies, atomic clusters)?
7) In Fig.4b and Fig.4c we can see beautiful structures made by atom manipulation but their electronic states signatures are missing. It is important to compare how similar or different they are versus previous reports on In/InAs(111)A [PRL 115, 076803 (2015)] and Cs/InSb(110) [Science 380, 1048-1052 (2023)] for validation of this platform.
8) In the last part the authors present stability of assembled structures (5 atoms) up to 44 K and individual Cs atoms up to 77 K. What is the reason for this difference in diffusion barrier? Does this effect scale with structure size?

---

## Round 2 · Referee Report · Anonymous (Referee 1) · 2024-1-22

Report

The manuscript resubmitted by the authors has been improved quite a lot and I am happy to recommend publication in SciPost Physics. I have one remaining question though and it would be nice if the authors could add this info to their manuscript:
It is valuable to see spectroscopy added to Figure 4. I am just puzzled by the way the FWHM of the peak is 30mV, which is still kind of wide for a gapped system. Can the authors compare it to other platforms like Cs/InSb (Science 380, 2023, Radboud group, FWHM = 7mV) or another recent paper on a noble metal 2DEG (1-5mV, Hamburg group, Nature 621, 2023)? Even the Crommie corral from 1993 seems to have line widths on the order of 30mV if I remember correctly. What is the physical reason for the line width in InAs, because as the authors write, there are no bulk states to scatter into? I would recommend adding this discussion to the paper and citing these other works for comparison.

---

## Round 2 · Referee Report · Philip Moriarty (Referee 2) · 2024-2-4

Report

The authors have systematically and carefully addressed each of the comments/queries in my previous review. I am happy to recommend publication of the manuscript in its present form.

However, I agree with another referee on the matter of the linewidth of the bound state. It would be very helpful if the authors could add a brief note on the linewidth issue to their manuscript.

---

## Round 2 · Author Response

Dear editor,

We thank the reviewers for their careful reading of our manuscript and the constructive feedback. This has helped us to improve our manuscript. We hereby submit a revised version.

Under the list of changes, you will find a detailed point-by-point response to all the questions, comments and remarks of all three reviewers.

We hope that our rebuttal, together with the revised version of the manuscript addresses the concerns of the reviewers adequately. We look forward to hearing from you.

On behalf of all authors,
Ingmar Swart

---

## Round 2 · List of Changes

Report 4
Referee: Ligthart et al. present a study on single Cs atom manipulation on semiconducting InSb(111)A surface including different manipulation modes. The paper is well written, clear for the reader and its strength lays in detailed description of lateral atom manipulation of adatoms on a semiconductor. I can recommend publication of this work after addressing the points listed below.
Response: We thank the reviewer for her/his appreciation of our work.

Referee point 1: In the introduction authors state that the use of semiconductors as a substrate for creating artificial electronic states allows for higher energy resolution in STS. Do authors mean to say that use of a substrate directly influences STS resolution? Addressing the physical reason for sharper energy states obtained for atom assembled quantum dots on semiconducting substrates would be beneficial for clarifying the above statement.

Response: We agree with the referee that this point can be clarified. Artificial lattices constructed using CO/Cu(111), make use of the 2D electron gas (surface state) of the Cu(111) substrate. However, CO molecules on the surface can also scatter the surface state electrons to bulk states, thereby reducing the lifetime. In the case of semiconductors, the charged adatoms induce states in the band gap of the semiconductor. Consequently, there are no bulk states to scatter to.

Action: On page 2, the sentence “An advantage of using semiconductors over metallic substrates is that the former allow for higher energy resolution in scanning tunnelling spectroscopy experiments.” was adjusted and the following sentences was added.
“An advantage of using semiconductors over metallic substrates is that the former allow for higher energy resolution of the electronic states of the artificial lattices. Lattices constructed using CO/Cu(111) make use of the 2D electron gas (surface state) of the Cu(111) substrate. However, CO molecules on the surface can also scatter the surface state electrons to bulk states of the Cu(111), thereby reducing the lifetime and causing significant broadening of the confined states .\cite{Freeney2020} In the case of semiconductors, the charged adatoms induce states in the band gap of the semiconductor due to local band bending. Consequently, there are no bulk states available to scatter to which leads to much narrower linewidths for confined states on semiconductors compared to metals. However, because of the more corrugated potential energy landscape, atomically precise manipulations on semiconductor surfaces are more demanding than on metallic substrates. For successful lateral manipulation the tip-adsorbate interaction needs to be stronger than the adsorbate-substrate interaction and the barrier for lateral motion should be smaller than for desorption.”

Referee point 2: Further in the introduction authors states that atom manipulation is more demanding on III-V semiconductor due to its polar and covalent character and I am missing a link. Can the authors elaborate on how exactly this character influences atom manipulation?

Response: The referee is correct that this point could be described more clearly. Metals have a relatively flat potential energy landscape whilst the potential landscape for III-V semiconductors is more corrugated. For lateral manipulation, the energy barrier for lateral motion should be smaller than for desorption. Furthermore, the bond between tip and adsorbate needs to be stronger than between adsorbate and substrate. Due to the corrugated potential landscape these conditions are easier to achieve on metals than on semiconductors.

Action: On page 2, the second paragraph was adapted as discussed under point 1.

Referee point 3: As a part of a motivation a part addressing the use of Cs atoms is missing. InAs(111)A substrate has native In adatoms which were previously used by e.g. Fölsch et al. [PRL 103, 096104 (2009) or Nature Nanotech 9, 505–508 (2014)]. What is the benefit of using Cs?

Response: Indeed, (native) In adatoms can also be used to construct lattices. The advantage of using Cs is that, as we show in our manuscript, they can also be manipulated laterally (not possible with In). Being able to use multiple manipulation techniques makes it easier to construct artificial lattices. The reason we chose Cs is that it is larger (making it easier to push/pull over the surface), while having the same charge as In.

Action: On page 2, the following sentence was added to the final paragraph of the introduction.
“However, the larger radius of the Cs compared to the substrate atoms is expected to facilitate lateral manipulation.”

Referee point 4: The vertical manipulation part of the paper is similar to the approach already described previously for In adatoms on the same substrate by Fölsch et al. [PRL 103, 096104 (2009)] but for with much less details. It would be nice to see a similar evaluation for Cs atom manipulation presented here especially that the lateral manipulation description is more detailed.

Response: As pointed out by the referee, the vertical manipulation procedure for Cs and In are very similar. Therefore, we did not investigate the vertical manipulation procedure in detail but rather focussed on the lateral manipulation method, which is not possible for In.

Action: -

Referee point 5: The authors state that the success rate for vertical manipulation “will vary somewhat” for different tip apexes based on data obtained for only single tip apex. What is the tip apex termination in this particular case? How does manipulation compare between bare PtIr, In terminated and Cs terminated tips?

Response: Like in many STM experiments, we cannot be sure of the precise tip-termination. In our experiments, we first created an In terminated tip, following a procedure developed by Fölsch. Furthermore, it is likely that the tip-apex is covered by some Cs atoms (intentional and unintentional pick up). During tip-preparation, we often deposited In and sometimes Cs atoms on the surface.

Action: An additional sentence about the tip preparation was added to the manuscript on page 3 in the Method section.
“The tip was coated with In using the following procedure. First, the tip was brought close to the surface (typical parameters: 500 mV, 300 pA). Then, we turned off the feedback loop and applied a bias of 10 V for a few seconds. This locally melted the surface, causing the formation of an In cluster. Small tip indentations in these clusters result in an In coated PtIr tip.”

Referee point 6: The difference for data obtained for push and pull traces is not clear (Fig.3d). Are they obtained in different crystallographic directions or with different tip terminations? Is there any influence for manipulation based on local chemical environment (vacancies, atomic clusters)?

Response: We agree with the referee that this point can be explained more clearly. We sometimes observed pushing behaviour along a particular direction. Interestingly, when moving the atom back to its original position (along the inverse trajectory), it was pulled. We therefore tentatively think that the method of manipulation depends on the mesoscopic details of the tip (shape, chemical composition, etc.). We did not observe any influence from the local chemical environment, but most manipulations were done on the unperturbed InAs(111)A surface.

Action: On page 7, the following sentence was added.
“Interestingly, the same tip can exhibit pushing and pulling behaviour when moving in opposite directions. We observed pushing characteristics along a crystallographic direction. Upon moving the same atom back to its original position along the same trajectory using the same tip, pulling motion was observed. This suggests that the mesoscopic tip shape influences the pulling or pushing behaviour of the tip.”

Referee point 7: In Fig.4b and Fig.4c we can see beautiful structures made by atom manipulation but their electronic states signatures are missing. It is important to compare how similar or different they are versus previous reports on In/InAs(111)A [PRL 115, 076803 (2015)] and Cs/InSb(110) [Science 380, 1048-1052 (2023)] for validation of this platform.

Response: We agree that it is important to assess the similarities and differences in artificial lattices made with In and Cs atoms. However, we feel that this discussion falls outside of the scope of the present manuscript, which focusses on manipulation protocols. A proper discussion of similarities and differences would (1) cause the manuscript to lose focus and (2) to approximately double in length. We do agree that it is important to show at least the confined electronic states of the structures built, to show that this system can be used to create artificial lattices. We therefore included the confined electronic state created by the Cs atoms, but left out the full comparison of the two systems.

Action: An extra paragraph was added on the bottom of page 7 together with a new Figure (4d) to show the confined state in the hexagon of Cs atoms.
“Bias spectroscopy was performed inside the hexagon shown in Figure 4c. Figure 4d shows the dI/dV curves acquired inside the hexagon and on the clean InAs(111)A. The spectrum of the hexagon exhibits a sharp peak at -0.25V, demonstrating the formation of a localized electronic state inside the bandgap of the substrate. This verifies that the Cs atoms on InAs(111)A are charged. The full-width-at-half-maximum of the peak is 33 mV, much narrower than lineshapes observed for quantum corrals of CO/Cu(111).\cite{Freeney2020} This demonstrates the feasibility of constructing artificial lattices using the Cs on InAs(111)A platform.”

Referee point 8: In the last part the authors present stability of assembled structures (5 atoms) up to 44 K and individual Cs atoms up to 77 K. What is the reason for this difference in diffusion barrier? Does this effect scale with structure size?

Response: We agree with the referee that this point can be clarified. The structures were stable to at least 45 K. Due to an unintended tip-crash, we do not have experimental data on the stability of assembled structures at higher temperatures. However, we did find that individual Cs atoms were stable at T = 77 K. Hence it is likely that structures are stable at temperatures above 45 K, but this needs further study. It is clear that our text was insufficiently clear, and we have made the changes indicated below.

Action: In the introduction the last sentence “The structures remained stable until a temperature of at least 44 K and at 77 K no hopping of individual Cs atoms was observed.” was adjusted to the following: “The structures were observed to be stable until a temperature of at least 44 K. Due to an unintentional tip-crash, we do not have data for higher temperatures. However, individual Cs atoms remained stable at 77 K, suggesting a higher stability for artificial structures.”Furthermore, on page 8, the he following sentence was adjusted. From: “Upon further heating, the tip crashed. Figure 5b and c show STM topographs of the same area, taken at T = 77 K. Individual Cs atoms did not show thermally induced diffusion, an image of the Cs atoms was taken, with a time difference of 33 hours. This suggests that manipulations can also be performed at higher temperatures.” To: “Upon further heating, the tip crashed and the stability of the structures above T = 44 K could not be studied further. Figure 5b and c show STM topographs of the same area, taken 33 hours apart. The sample temperature was T = 77 K. It is clear that at this temperature, there was no significant diffusion of Cs adatoms. This suggests (1) that artificial lattices could be assembled at this temperature and (2) that the structures could be stable at temperatures above T = 44 K, but further study is needed.” 

Report 3
Referee:
Strengths
1. Clearly written paper on a subject of particular topical interest.
2. Meets SciPost acceptance criterion: “Open a new pathway in an existing or a new research direction, with clear potential for multipronged follow-up work.”
3. Provides new protocols for single atom manipulation on an important substrate/platform .
Weaknesses
1. Some claims re. manipulation mechanisms – and, as pointed out by Reviewer #1, re. the charge state of the Cs adatoms – do not seem to be fully supported by experimental evidence. See comments in report below.
2. Not clear how the claimed indium termination of the tip is maintained throughout the experiments.
3. Although the experimental measurements are generally described in more than adequate detail, in some places greater detail on error bars/uncertainties could be provided.

Report
This is an important and well-written paper on a topic of considerable interest to the nanoscience, surface science, and condensed matter physics communities: single atom manipulation on semiconductor surfaces. The Cs/III-V system is likely to receive renewed attention following the elegant and pioneering “quantum simulator” paper from the Radboud group (Khajetoorians et al.) published earlier this year (Science 380 1048 (2023)). Previous similarly elegant work by Folsch et al. [e.g. PRL 115, 076803 (2015)] on the assembly of quantum dot molecules on InAs(111)A has demonstrated that this particular surface is especially amenable to controlled single atom positioning with an STM tip.
As such, this paper is particularly timely and establishes protocols for manipulation of Cs atoms on InAs(111)A. I agree with the points raised by the first reviewer (i.e. Anonymous Report 1 on 2023-10-27) and would also ask for the following points to be addressed in a revised version of the manuscript:

Response: We thank Prof. Moriarty for his response and the positive reception of our work.

Referee point 1a: The Introduction draws a distinction between vertical manipulation methods that transfer an atom from the tip to the substrate and vice versa, generalising that these mechanisms involve the electric field and inelastic tunnelling for surface-to-tip transfer, and vdW forces/short-range forces for tip-to-surface transfer. I think this is a rather strong statement if it is meant in a general sense. Are the authors suggesting that these mechanisms hold for the InAs(111)A substrate or do they mean that *all* vertical manipulation events, i.e. for any tip-sample combination, can be explained in this manner?

Response: We did not want to claim that all vertical manipulation events can be attributed to the same effects. We have modified the sentences referred to by Prof. Moriarty.

Action: The following sentence was adjusted.
“The transfer of the adatom from the surface to the tip is mediated by a tip-induced electric field and vibrations that arise due to inelastically tunneling electrons. The transfer from the tip towards the surface is dependent on short-range forces such as Van der Waals forces and is achieved by an almost point contact between tip and sample.”
to :
“For In atoms on In/As(111), this transfer of the adatom from the surface to the tip is mediated by a tip-induced electric field and vibrations that arise due to inelastically tunneling electrons. The transfer from the tip towards the surface is in this case dependent on short-range forces such as Van der Waals forces and is achieved by an almost point contact between tip and sample. \cite{Yang2012}”
Referee point 1b: In a similar vein to 1(a) above, on p. 4 the authors argue that the jump of the adatom is "caused by inelastically tunnelling electrons which excite vibrations [20]." The reference is to a detailed analysis by Fölsch's team that demonstrates that this is the case for In adatoms on InAs(111)A. It is not immediately obvious that the inelastic tunnelling mechanism will also dominate for the transfer of Cs adatoms. Do the authors have direct evidence for this inelastic tunnelling mechanism in their experiments on Cs/InAs(111)A?

Response: This sentence was meant to convey that, given the similarities in charge and adsorption site, it is likely that the pick-up mechanism for In and Cs atoms is similar. However, we do not have direct experimental proof that inelastic tunnelling is responsible for picking up Cs atoms. We therefore modified the sentence.

Action: The sentence on page 5 “The jump of the adatom is caused by inelastically tunneling electrons which excite vibrations.\cite{Yang2012} Hence the likelihood of picking up an adatom is larger at higher tunneling currents.” was adjusted to the following. “The vertical manipulation procedure for Cs atoms on InAs(111)A, as well as the experimental signatures, are similar to the case of In adatoms. This suggests that the manipulation mechanisms are similar for the two different atoms, i.e. that jump of the Cs adatom is caused by inelastically tunnelling electrons which excite vibrations.\cite{Yang2012}. This is consistent with the observation that the likelihood of picking up an adatom is larger at higher tunnelling currents.”

Referee point 1c: The Introduction is a single, long paragraph. I suggest breaking this up into at least two shorter paragraphs to aid readability and clarity.

Response: We thank the referee for this suggestion.

Action: We have split the introduction into 4 paragraphs.

Referee point 2: The final sentence of the Introduction section is intriguing: "The structures remained stable until a temperature of at least 44 K, and at 77 K no hopping of individual Cs atoms was observed." This suggests that the barrier for diffusion for Cs atoms that comprise an STM-assembled structure is lower than that for individual Cs atoms. Am I interpreting this correctly?

Response: Referee 4 also raised this point (see point 8 above). Our text was insufficiently clear. We refer to our comment there.

Referee point 3: This point relates to the first reviewer's questions re. tip structure. In the second sentence of the Methods section, the authors state that an "In-coated PtIr tip was used for *all measurements*." (Emphasis mine.) How do the authors know that the tip is In-terminated, and, equally importantly, that it remains In-terminated throughout the manipulation experiments and spectroscopic probing? Is there not a finite (and perhaps rather large) probability of Cs being picked up by the tip?

Response: Like in many STM experiments, we cannot be completely sure of the precise tip-termination. In our experiments, we first created an In terminated tip, following a procedure developed by Fölsch (now described in the methods section). We agree with the referee that it is likely that the tip-apex is covered by some Cs atoms (intentional and unintentional pick up). During tip-preparation, we often deposited In and sometimes Cs atoms on the surface.

Action: We modified the following sentence about the In-coated PtIr tip.
“An In-coated PtIr tip was used for all measurements.” became “An In-coated PtIr tip, possibly coated by some Cs atoms, was used for the measurements.” Furthermore, in the methods section, we added the following text about the tip preparation. “The tip was coated with In using the following procedure. First, the tip was brought close to the surface (typical parameters: 500 mV, 300 pA). Then, we turned off the feedback loop and applied a bias of 10 V for a few seconds. This locally melted the surface, causing the formation of an In cluster. Small tip indentations in these clusters result in an In coated PtIr tip.”

Referee point 4: p.2, third line from the bottom: "...reflection high energy electron diffraction (RHEED)..."

Action: We thank the referee for pointing out this mistake. We have corrected it.

Referee point 5: In Section 3.1, I suggest moving the sentence that starts "The In atoms are naturally present on the InAs(111)A surface..." so that it's the second sentence of that paragraph.

Action: We implemented this suggestion.

Referee point 6: First sentence in main text on p.4: "...results in a sudden jump down in the current". This should be "sudden increase in the current".

Action: We implemented this suggestion, and we assumed that the referee means a decrease in the current.

Referee point 7: p.5, second sentence. "Putting the adatom down is not induced by the electric field..." How do the authors rule out the influence of the electric field?

Response: Transferring the Cs atom back to the surface was possible at both positive and negative voltages. It could even be done without applying a bias.

Action: On page 5, we modified the sentence to “As Cs atoms could be transferred back to the surface at various positive and negative voltages, the deposition process is not induced by the electric field, but is mediated by the near point contact between tip and adatom.”

Referee point 8: Fig.3 -- The mean hopping distances in Fig. 3(d) are quoted to three significant figures (i.e. down to the single pm level). What's the standard error on these mean values?

Response: We did not investigate the standard deviation systematically, but we expect it to be on the order of 10 pm. We therefore removed one digit.

Action: One digit was removed from Figure 4.

Referee point 9: For Figure 4 (and in line with Reviewer 1's comment re. error bars), what was the total number of manipulation events used to generate the histogram? (My apologies if this information is in the paper and I missed it.) Is there a bias polarity dependence?

Response: In addition to Reviewer 1’s comment on the error bars (response written below), the total number of manipulation events was 120, wherefrom the most at 300pA current setpoint (80). The referee is right that it is useful to add the total number of manipulations events.

Action: The following sentence was adapted to the referee’s comment. The sentence about the success rate at 300pA was removed as it became abundant. “We have investigated the reliability of the lateral manipulation by recording the success rate as a function of setpoint current, using one tip apex.” Became “We have investigated the reliability of the lateral manipulation by recording the success rate as a function of setpoint current, using one tip apex 120 manipulations were performed.”

Referee point 10: Caption to Fig. 5, final sentence. I assume this should be "900 mV and 30 pA".

Action: We thank the referee for noticing and we have corrected the error.

Report 1:
Referee: Ligthart et al. present a study of a novel platform for artificial lattices consisting of individual Cs atoms on InAs(111)A. They report on different successful manipulation modes with the STM tip and present examples of small artificially constructed objects, including densely packed structures of Cs atoms. The topic is timely, the paper is well written and this is indeed an interesting development, so I would clearly recommend publication of this work after the authors commented on my questions below. Most importantly, I would like to see a little more data backing the claim that this platform is interesting for studies of artificial lattices.

Response: We thank the referee for his/her appreciation of our work and his/her thorough response. We agree that the suitability to use this platform to construct artificial lattices could be highlighted more. We therefore added an additional figure showing that the closed-packed hexagon indeed features a confined state (details given below). However, the aim of the present manuscript was to focus on the manipulation protocols. A full comparison of this platform with other platforms would cause the present text to lose focus.

Referee point 1: The authors present a nice statistical evaluation of the manipulation's success rate in Fig. 4a (please add statistical error bars to Fig. 4a though). It would be interesting if they could provide similar statistics on the vertical manipulation mode.

Response: The success rate for lateral manipulation is dependent on tip termination. Some tips do not manipulate. If we could not manipulate Cs atoms, we applied voltage pulses until we had a tip that would manipulate. Hence, it is difficult to provide meaningful error bars on the numbers in Figure 4a.
The procedure for vertical manipulation of Cs is very similar to that of In, which has been studied in detail \cite{Yang2012}. Therefore, we did not investigate the vertical manipulation procedure in detail but rather focussed on the lateral manipulation method, which is not possible for In on InAs(111)A.

Action: At the end of section 3.2, we added a sentence to explain why we did not study the vertical manipulation in detail:
“Since the vertical manipulation procedures for Cs and In on InAs(111)A are similar, and the latter has been investigated in detail \cite{Yang2012}, we did not investigate this method further.”

Referee point 2: On page 4, they state that stable tips does not change significantly after vertical manipulation, however, the tip in Fig. 2b-d becomes unstable after dropping the Cs atom. Do they have other examples for vertical manipulation with no clear tip changes?

Response: Indeed, signatures of minor tip-instabilities are present in 2d. We chose to show this image, as one can clearly observe the vacancy reconstruction of the substrate as well as the characteristic shapes of the In and Cs atoms. This facilitates the comparison of the images. We have other examples, swhich can be provided upon request (The SciPost platform does not allow uploading images in a rebuttal).

Referee point 3: In general, it would be interesting if the authors could comment on how they prepare the tip apex. Can you fix an unstable tip directly on the InAs surface? Are there band bending effects for different tip apexes?

Response: The referee is correct that this points could be elaborated on. The other two referees also commented on this aspect. We follow a tip-preparation protocol developed by Fölsch. We added this information to the methods section. It is indeed possible to fix unstable tips directly on the InAs(111)A surface using this approach.

Action: An extra sentence was added on page 3 in the Method section. “The tip was coated with In using the following procedure. First, the tip was brought close to the surface (typical parameters: 500 mV, 300 pA). Then, we turned off the feedback loop and applied a bias of 10 V for a few seconds. This locally melted the surface, causing the formation of an In cluster. Small tip indentations in these clusters result in an In coated PtIr tip.”

Referee point 4: In Fig. 3, it is shown that both pulling and pushing mode can be achieved. I am puzzled though what the difference is: does it mainly depend on the precise tip apex? Or is there a dependence on (e.g.) directionality of the lateral manipulation? Please clarify.

Response: We sometimes observed pushing behaviour along a particular direction. Interestingly, when moving the atom back to its original position, it was pulled. We therefore tentatively think that the method of manipulation depends on the details of the mesoscopic details of the tip (shape, chemical composition, etc.)

Action: On page 7, the following sentence was added.
“Interestingly, the same tip can exhibit pushing and pulling behaviour when moving in opposite directions. We observed pushing characteristics along a crystallographic direction. Upon moving the same atom back to its original position along the same trajectory using the same tip, pulling motion was observed. This suggests that the mesoscopic tip shape influences the pulling or pushing behaviour of the tip.”

Referee point 5: The authors write that semiconductors "allow for higher energy resolution in scanning tunnelling spectroscopy". I know what they want to express, but please rephrase this because it sounds as if the actual energy resolution improved, which is not true. Please also add one or two sentences about the physical reason for this improved "effective" resolution and the advantage of their platform over other experimental platforms for artificial lattices. This would highlight the work's quality and relevance throughout the paper.

Response: We agree with the referee that this point can be improved.

Action: On page 2, the sentence “An advantage of using semiconductors over metallic substrates is that the former allow for higher energy resolution in scanning tunnelling spectroscopy experiments.” was adjusted and the following sentences was added.
“An advantage of using semiconductors over metallic substrates is that the former allow for higher energy resolution of the electronic states of the artificial lattices. Lattices constructed using CO/Cu(111) make use of the 2D electron gas (surface state) of the Cu(111) substrate. However, CO molecules on the surface can also scatter the surface state electrons to bulk states of the Cu(111), thereby reducing the lifetime and causing significant broadening of the confined states .\cite{Freeney2020} In the case of semiconductors, the charged adatoms induce states in the band gap of the semiconductor due to local band bending. Consequently, there are no bulk states available to scatter to which leads to much narrower linewidths for confined states on semiconductors compared to metals.”

Referee point 6: They write: "Given the low ionization energy of Cs, and its behaviour on InSb(110), the Cs atoms on InAs(111)A are charged". I don't quite understand if this is a speculation or a backed claim? Please clarify.

Response: We thank the referee for pointing this out. The new figure 4d shows that the 7 Cs atoms forming a hexagon leads to a confined state in the band gap of the substrate. This only happens when the atoms are charged (the confined state are formed by local band bending). Hence, this observation supports the claim that the Cs atoms on InAs(111)A are charged.

Action: The sentence above on page 3 was adapted. From: “Given the low ionization energy of Cs, and its behaviour on InSb(110), the Cs atoms on InAs(111)A are charged.” To: “Given the low ionization energy of Cs, and its behaviour on InSb(110), the Cs atoms on InAs(111)A are expected to be charged. This is confirmed by the observation of a localized electronic state for a cluster of Cs atoms (see below).” In addition, on page 7, we added the a sentence relating the observation of a confined electronic state to the charge of the Cs atoms (also see next point).

Referee point 7: Now that the authors showed that Cs can be moved, it would be important to provide more data showing that this platform actually qualifies for artificial lattice studies. For instance, is there any (sharp?) resonance peak around the structures presented in Fig. 4 verifying the confinement of charges?

Response: The referee is correct that it is important to provide data to show that Cs on InAs(111)A can be used to construct artificial lattices. A prerequisite for this is the formation on a confined electronic state. We therefore added a figure to that demonstrates that such a sharp resonance peak is indeed induced by the closed packed hexagon. We feel that a full comparison between different platforms, although important and relevant, falls outside the scope of the present manuscript, which focusses on the manipulation procedures of Cs on InAs(111)A. We therefore prefer to reserve those results for another manuscript

Action: An extra paragraph was added on the bottom of page 7 together with a new Figure (4d) to show the confined state due to the Cs adsorbates. “Bias spectroscopy was performed inside the hexagon shown in Figure 4c. Figure 4d shows the dI/dV curves acquired inside the hexagon and on the clean InAs(111)A. The spectrum of the hexagon exhibits a sharp peak at -0.25V, demonstrating the formation of a localized electronic state inside the bandgap of the substrate. This verifies that the Cs atoms on InAs(111)A are charged. The full-width-at-half-maximum of the peak is 33 mV, much narrower than lineshapes observed for quantum corrals of CO/Cu(111).\cite{Freeney2020} This demonstrates the feasibility of constructing artificial lattices using the Cs on InAs(111)A platform.”

---

## Round 3 · Author Response

Dear editor,

We thank the referees for their input and you for giving us the opportunity to resubmit our manuscript.

As outlined below, we have included a brief discussion on the linewidths we observe in our experiments in our revised manuscript. As suggested by the referee, we compare the linewidth to those observed for other non-superconducting material systems. When making this comparison, it is important to realize that linewidths depend strongly on the density of adsorbates on the surface, as well as on barrier height. For example, for In on InAs(111)A, the FWHM varies between ~30 mV for a linear chain of 22 atoms, to ~10 mV for a chain of 6 atoms. Similarly, for corrals on Cu(111), the peak width increases as the corrals become smaller (more scatterers per unit area). In addition, the peaks become wider the higher they are in energy. A detailed investigation of how linewidth for Cs/InAs(111)A depend on nanostructure shape and size is certainly very interesting and worth undertaking but we feel that this is outside the scope of the present manuscript, which focusses on the manipulation of adsorbates.

We hope that the changes described below sufficiently address the final comments of the referees.

On behalf of all authors,

Best regards from Utrecht,
Ingmar Swart

---

## Round 3 · List of Changes

Following the suggestion of the referees, we modified the text discussing the full-width-at-half-maximum on page 8.
Form version 2:
“The spectrum of the hexagon exhibits a sharp peak at -0.25V, demonstrating the formation of a localized electronic state inside the bandgap of the substrate. This verifies that the Cs atoms on InAs(111)A are charged. The full-width-at-half-maximum of the peak is 33 mV, much narrower than lineshapes observed for quantum corrals of CO/Cu(111). [6] This demonstrates the feasibility of constructing artificial lattices using the Cs on InAs(111)A platform"

To version 3:
"The spectrum of the hexagon exhibits a sharp peak at -0.25V, demonstrating the formation of a localized electronic state inside the bandgap of the substrate. This verifies that the Cs atoms on InAs(111)A are charged and demonstrates that this platform can be used to construct artificial lattices. The full-width-at-half-maximum (FWHM) of the peak is 33 mV, much narrower than lineshapes observed for artificial lattices made using the CO/Cu(111) platform, but comparable to that of a 22-atom long In chain on InAs(111)A. [15,30] For In/InAs(111)A, the linewidth depends on the density of In atoms in the nanostructure (also expected for Cs/InAs(111)A) and can be as small as 10 mV for a line of six In atoms. [30] This is similar to the ≈7mV FWHM observed for the Cs octagon on InSb(110). [7]"

---

## Editorial Decision

published